# Does Environmental Change Affect Migration Especially into the EU?

Dina Moawad [1,2]

1 Department of Business and Economics, University of Naples Parthenope, 80132 Naples, Italy; dina.moawad@studenti.uniparthenope.it
2 Faculty of Social Science & Public Policy, University of King's College London, London WC2B 4BG, UK

**Abstract:** Environmental shock migration is a pressing phenomenon that became prominent with the continuous emergence of natural disasters and climatic shocks worldwide. In order to cope with these various disasters or shocks, people choose to migrate either internally, internationally, permanently, or temporarily; the paper named this phenomenon "environmental shock migration". For a holistic understanding, this paper analyzes the impact of environmental changes on migration and discusses the relevant consequences, specifically in the EU region. The paper demonstrates that natural disasters and climatic shocks as environmental changes lead to several forms of shock migration and differ depending upon the context of migration, the duration, the number of migrants, and the region. A comprehensive literature review will be provided to tackle the work of previous scholars and identify the gaps required to be studied in the future.

**Keywords:** shock migration; environmental change; natural disasters; climate change; EU





## 1. Introduction

People tend to move when natural or climatic shocks happen, which is known as "Environmental shock migration". It is a critical phenomenon that occurs regularly, especially with recent shocks consistent with climate change. The significance of studying environmental shock migration lies in its dynamic nature, which is constantly changing and evolving. Since the consequences of this shock migration have extended to several countries, deteriorating the economic conditions, understanding this phenomenon and how to deal with it became an essential matter for several scholars and policymakers.

Generally, environmental changes occur due to natural disasters such as earthquakes, hurricanes, fires, floods, and tsunamis. Therefore, early actions are needed in the advanced stages. Several countries have been prone to shock migration; for instance, Haiti and Pakistan faced severe earthquakes, which led to massive numbers of internally displaced people. Afghanistan has also undergone severe floods and droughts since 2018, which has led to large numbers of people evacuating their homes. The Internal Displacement Monitoring Centre (IDMC) has been evaluating such connections for more than 20 years. Throughout these years, natural disasters have been the primary reason behind displacement. Numbers showed that 23.7 million people were displaced in 2021 alone due to natural disasters. It is anticipated that around 200 million people will evacuate their houses by 2050 due to environmental changes (Myers 2002).

More appropriate, reliable data are needed to analyze the phenomenon of environmental migration in Europe. It is hard to identify if this migration in Europe was due to environmental events or other factors. The lack of information regarding environmental migration in Europe has caused environmental challenges to be widely neglected (Jäger et al. 2009). However, it has been proven that environmental changes are already affecting the situation in Europe. In 2003, around 70,000 people died from the heat wave across Europe (IDMC). A study (Missirian and Schlenker 2017) also showed a strong connection between changes in temperature and movement application in Europe.

According to the IDMC (2017), natural disasters have displaced more than 66,000 people in Europe. Such a number is small compared to the global scale. However, it still widely affects European communities, such as the wildfire in Corsica, France, which led to the movement of 10,000 people in 2017 (IDMC 2017). The indicators of environmental change in Europe are evident in the increase in temperatures, rainfall variability, changes to the ecosystem, and wildfires that divide the European continent. The southern and Mediterranean regions have a shrinking water supply, wildfires, and crop production. The northern part is more subject to ecosystem changes and floods. Eastern and central Europe are likely to witness an increasing rate of fires and a decline in rainfall, affecting the water supply (IPCC 2014).

The relationship between natural disasters, climate change, and migration is significant and needs to be studied, especially in the EU and its small regions that are the most vulnerable but with insufficient resources to deal with this repetitive random change (Kahn 2005; Tol et al. 2004; Mendelsohn et al. 2006). Some of the prior studies showed that the places most exposed to repeated environmental change have a less marginal loss, but the adaptation policy to deal with these events is costly and needs time to become efficient (Hsiang and Narita 2012). Natural disasters and climate change affect agricultural output, as shown by the literature, and migration is one of the responses of households and indirect consequences of this effect, especially in developing regions that have mainly depended on agriculture until now, as in some African countries (Mendelsohn and Dinar 2009). Previous work in this area mentioned that environmental change leads to people's internal displacement and to their seeking new stable and safe places to live temporarily or permanently (Barrios et al. 2006; Mastrorillo et al. 2016).

This paper focuses on the relationship between environmental shocks and migration, especially in Europe, considering natural disasters and climatic shocks as events that lead to environmental changes. It mentions the definitions, significant drivers, and types of environmentally induced migration. It also identifies some gaps in the literature and essential topics to be examined in future studies. The paper is divided as follows: background, research methodologies, definitions, the relationship between natural disasters, climate change, and migration, mediating factors and environmental shock migration, types of environmental shock migration, a summary of gaps in the literature, required future work, and policy recommendations, especially in the EU.

## 2. Background

Researchers expected many people to reallocate their living places because of natural disasters or climate change, establishing a new concept of "Environmental Migration". (Houghton et al. 1992). The empirical work of environmentally induced migration started not from the economic side but from geographic and other social sciences. The survey (Roncoli et al. 2001) in Burkina Faso in the year after its drought in 1997 provided a clear example of such studies. Demographic studies started to investigate the environmentally induced migration and climate shock effects as in the studies by (Barrios et al. 2006; Marchiori et al. 2012; Stern 2007), which opened a door to the perception that many countries would suffer from adverse consequences of natural disasters and climate change. The first UN intergovernmental report highlighted that human displacement is one of the most significant consequences of natural disasters and climate change, especially in developing countries. (Piguet et al. 2011) also investigated the relationship between natural disasters, climate change, and migration in an overview of the essential elements with previous studies. The authors concluded that natural disasters and climate change are united in leading to worse consequences; they happen faster than expected and have a more dramatic effect on developing countries.

The World Bank (Gray and Wise 2016) delivered the first cross-country analyses to identify the environmentally induced migration in Africa for five countries (Burkina Faso, Kenya, Nigeria, Senegal, and Uganda). A significant contribution was that they collected and unified all empirical approaches and climate change definitions across these

five countries to understand the relationship between natural disasters, climate change, and migration. The results showed no robust relationship between rainfall variability and migration, while the temperature may affect migration decisions by different scales from country to country. For example, temperature variability positively affects migration patterns in Uganda, while vice versa in Kenya and Burkina Faso.

Despite this work in Africa, a gap in the empirical body recently started to be filled to prove this relationship (Nordås and Gleditsch 2007; Scheffran and Battaglini 2010). On the one hand, some found no relationship between natural disasters, climate change, and migration in the short or long run. In contrast, others found no relation in the short run, while in the long run, it is negative or positive. (Di Falco et al. 2012) also showed a causal relationship between natural disasters, climate change, and migration, as it claimed that changes in the temperatures and rainfall affected people's productivity, leading to migration. It confirmed that migration was a response to the decline in agricultural productivity caused by environmental changes, especially in developing countries.

Natural disasters and climatic shocks lead to several forms of migration depending upon the context of migration, the duration, the number of migrants, and the region. (Gray and Mueller 2012) showed that moderate flooding in Bangladesh from 1994–2010 led to more people moving locally than internationally, especially the most vulnerable, such as women and low-income people. More studies went deeper into the relationship between displacement levels and the seriousness of an environmental event. For instance, in 2014, 32 massive environmental events led to 83% displacement, while only 17% was because of small and medium events (IDMC 2014). The studies also showed that from 2008–2014, 95% of global displacement due to natural disasters emerged from developing countries. Most of these displacements are middle-income people as they can afford migration and mostly do not have other options, while wealthy and low-income people are less likely to migrate after natural disasters. This is because rich people have coping strategies other than migration, while low-income people cannot afford migration.

In Nicaragua, (Kniveton et al. 2008) found that the hurricane changed rainfall intensity and affected the low-income rural households with insufficient resources to migrate more than the rich ones, so it also depends on the households' economic and social characteristics. The study also mentioned that the environmental change exposure levels are different across regions, so this hurricane led to a notable migration from more environmentally dependent regions such as the rural areas, and this was also confirmed later by (Carvajal and Pereira 2009).

(Baez et al. 2017) found that natural disasters lead to different implications. For instance, education and employment are the most affected aspects by floods. Tropical storms may lead to employment disability. Volcanoes affect the future capabilities of wealth accumulation. The study also showed that women exposed to natural disasters are more likely to affect their children's education than exposed fathers with extended consequences to several generations.

(Mbaye 2017) argued that post-natural disaster migration is for people who do not have any other adaption plan and can afford its costs. The study analyzed the pros and cons of post-natural disaster migration. The first advantage is that migrants will have new opportunities to cope after the disaster. Secondly, migrants transfer funds to their hometowns, relieving the shocking consequences. The disadvantages are apparent in the liquidity constraints, which directly affect low-income people who cannot afford migration. The study claimed that migration helps to relieve the implications of natural disasters while liquidity constraints constantly stand in the way of it. This is why low-income people do not migrate after slow-onset disasters, while they are forced to do so after fast-onset ones to the nearby regions.

(Pavel et al. 2018) showed that permanent and temporary shocks in Bangladesh most likely lead to migration, but the two types of shocks differ in their influence on the extent of migration. The study also confirmed that migration is a good coping strategy because the migrants' economic conditions are better after moving than those who did not move.

A meta-analysis (Hoffmann et al. 2020) investigated the effects of environmental changes on migratory patterns by combining findings from 30 country-level research. These studies jointly analyzed the global influence of both gradual and rapid onset events on migration. While this study agreed on the impact of environmental hazards on migration, the scale of this impact varies depending on the environment. The study showed that internal migration or mobility to low- and middle-income countries is more predominant, with a strong association in countries outside the Organization for Economic Co-operation and Development (OECD). Latin America, the Caribbean, and Sub-Saharan Africa have a robust correlation as middle-income agricultural countries. Surprisingly, the interaction between income levels and conflicts sometimes helps to mitigate and, to some extent, illuminate the complex link between environmental changes and migration. Combining these disparate migration responses with observable environmental modifications over the last few decades reveals possible hotspots of environmental migration. The technique used in the significant quantitative country-level studies plays a critical role in creating these discoveries, as shown by the 'Pub.', 'Period', 'Years', 'Region', and environmental elements addressed. This review provided a deeper awareness of the complicated interplay between environmental change and migration dynamics on a global scale by carefully investigating the heterogeneity in effect sizes across estimates and scrutinizing the mechanisms that may underpin variations in study results based on specific contexts and sample compositions.

(Beine and Jeusette 2021) presented a meta-analysis of the methodological approaches to the relationship between migration and climate change. The study explained the results' heterogeneity of previous empirical work by their methodologies which depend on many aspects. Firstly, regarding the relationship between climate change and migration, which could be positive or negative, the paper found that it could be direct if there is a significant causal relation or indirect if there is a channel or instrumental variable of climate change that affects migration. Secondly, regarding the climatic factors, the paper has divided them into two categories; the first category is long-term variables that affect migration slowly, such as temperature or precipitation levels, temperature precipitation variability, and soil moisture. The second category is the short-term or extreme variables such as extreme temperatures, extreme precipitations, floods, hurricanes, storms, and droughts. Thirdly, the paper found direct and indirect measures for the dependent variable. The direct measures are when the variable is extracted from the data directly, for example, when the migrant has been asked about his migration history/reasons, while the indirect measures are when the data are calculated from census data of migration or by using a proxy for mobility, such as the urbanization rate as a proxy for internal migration. Fourthly, for the channels, the four main channels that existed in the previous literature were the economic channel, agricultural channel, and urbanization channel, and if there is no specific channel, it will be an aggregate of channels.

(Huang et al. 2022) examined 957 survey samples from 12 disaster zones after the Wenchuan earthquake to understand the individuals' migration intentions and the factors influencing them. The study found that 45.2% are willing to migrate, indicating an instinctive desire for compensation and avoiding damage; however, this propensity has yet to result in a real decision. While the success of post-disaster restoration actions has given these places hope for the future, various variables continue to influence migratory decisions. The study used a dichotomous variable to understand these processes better, defining inhabitants' willingness to migrate as "move" or "stay" considering six contributing factors—gender, education level, family income, life convenience, future life expectations, and psychological feeling—to examine the migration decisions thoroughly. The study provided insights into residents' migration intentions and valuable policies for the living environment deterioration after earthquakes, which encourages residents in secondary earthquake disaster zones to migrate.

(Gray and Mueller 2023) considered Bangladesh as a case study to analyze people's ability to move after natural disasters. This study is based on five hypotheses, which are (1) people's long-term mobility increases due to natural disasters, (2) the consequences of

natural disasters are apparent in long-distance movements, (3) the most affected ones are the low-income people, (4) area-level shocks have more implications than household-level shocks, and (5) flooding and its induced displacement are more critical than crop stoppage. The study showed that natural disasters sometimes lead to long-term population mobility; this was distinct from the first hypothesis. Natural disasters also sometimes lead to long-distance movements as uncertainty and costs might accompany these movements, so internal movements are more likely to occur, which also affects the second hypothesis. The third hypothesis had conflicting results as the degree of the implications on the low-income and the rich people have yet to be proven. The fourth hypothesis was also rejected as it has been proven that area-level shocks are as crucial as household-level shocks. The study concluded that, unlike the overall assumptions, crop loss substantially affects mobility compared to floods.

Analyzing environmental migration in the EU shows that climatic shocks affect some push factors, leading to migration. Natural disasters and climatic shocks destroy infrastructure, houses, and sources of income, hindering production and trade and destroying capital (Baez and Santos 2008; Banerjee 2017; Cai et al. 2016; Gröger and Zylberberg 2016). For example, the Messina-Reggio Calabria earthquake in Italy destroyed the state's capital and massively affected people's living standards and wealth. Hence, their sources of income and financial safety were highly diminished, so staying in their home state was no longer an option (Spitzer et al. 2020). There is a debate on the effect of environmental shocks. On one side, these natural disasters lead to short-term migration and short distances (e.g., Gray and Mueller 2012; Gröger and Zylberberg 2016; Penning-Rowsell et al. 2013; Robalino et al. 2015). Conversely, some people might migrate long distances permanently (Boustan et al. 2012; Hornbeck 2012; Hornbeck and Naidu 2014).

Environmental change and its visible effects, which include a rise in extreme events such as fires, floods, storms, and earthquakes, have sparked increased alarm. (Gil et al. 2022) investigated the complex relationship between climatic phenomena and migration, in line with SDG 13, "Climate Action", in four Northern countries: Austria, Greece, Spain, and the United States. The study examined climatic events in these countries in 2021, delving into their economic, social, and cultural impacts, particularly regarding migratory trends. Despite recording migration data problems for this year, there is a clear tendency towards internal and short-term mobility.

The complicated interconnections between environmental change, human migration, and adaptation under the well-established governance structure of the EU were restricted and mostly security-focused. This shows limited comprehension of migration as a feasible means of adaptation to complex changes in the economic, social, political, demographic, and environmental sectors, whether internal or international. (Geddes and Jordan 2012) looked at the EU environmental policy, outlining its primarily used methods or instruments and identifying its fundamental dynamics within this policy sector. The literature demonstrated few intersections between environment and migration policies with a strong security focus, showing migration as a failure to adapt and an early response to possible disasters. The tools used to coordinate these policies needed to have hierarchical structures or clear bureaucratic standards and accepted practices. The lack of importance in staff training sessions and the absence of a mission statement such as the sixth article of the "Amsterdam Treaty" emphasized the topic's low profile. In contrast, the UK stated a separate strategy with an extensive research effort within its central 'Office for Science', indicating that comparable functional equivalents at the EU level needed to be more prominent. According to the Environmental Performance Index (EPI), the coordination of the topic inside the EU relied on a small group of coordinating instruments, primarily superficially.

To sum up, the previous literature about environmentally induced migration showed heterogeneous conclusions depending on different aspects of the phenomenon. The type and nature of the environmental shock play an essential role in formulating migration decisions. The geographical context also has a vital role, so the effect of these environmental

shocks in developing countries (Africa) is different from the developed ones (EU), as is explained in the upcoming sections.

For conducting the previous studies, varied methodologies have been used to tackle the environmentally induced migration phenomenon from all aspects, as illustrated in the following section.

### 3. Environmental Migration Methodologies

Multiple studies (Kniveton et al. 2009; Laczko and Aghazarm 2009; de Campos et al. 2017; Piguet 2010) explored environmental shock migration and the difference in its measuring methodologies. Different sources and data types lead to various migration methodologies. The data on natural disasters and climate change differ from the migration data, so combining them is challenging (Gray and Mueller 2023). Most of the previous literature worked in two main methodological categories: 1—The descriptive studies tried to determine the main shocks and disasters, the most vulnerable zones to these shocks, the effect of these shocks on its population, and the probability of future migration. 2—The analytical studies are more interested in explaining the causes of this migration and the primary reason, whether it is a natural hazard or something else.

(Beine and Jeusette 2021) firstly found that data frequency has an influential role in the estimation method of environmental change on migration. Using higher-frequency data means better explaining long-term displacements than short-term ones and using proxies of migration also leads to a better explanation of environmental shocks' effect on it. Secondly, the kind of regression method used is vital; for example, conditional regression better determines the environmentally induced migration relationship. Thirdly, the effect of environmental change on migration depends on specific conditions such as the importance of the agricultural side or the government-supporting policies, so using instrumental variables better explains this relationship. The study context regarding time or geography also affects the empirical method, so geographical context should be considered. For example, the analysis of the developing countries using the pooled logit method explains climate-induced migration better, which is compatible with the fact that developing countries are more vulnerable and have a double natural hazard (M. D. Cattaneo et al. 2019).

For better understanding, it is essential to explain the adapted definitions of the environmental shock migration phenomenon and the EU's position regarding it.

### 4. Definitions of Environmental Shock Migration

*Environmental shock migration* is defined as the abrupt movement of people due to a large-scale environmental change, so it is mostly an acute form of migration that differs from the planned one. Environmental change happens from natural disasters, which are the short-term events that happen unexpectedly, such as earthquakes, floods, tsunamis, and droughts, or climate shocks, which are long-term changes resulting from weather changes, such as temperature and rainfall changes over a long time.

The concept of "Environmental/Climate migrant" has been identified academically and politically. From an academic point of view, there was criticism for having a concept of environmental migrants because of the multi-causality problem, which means that migration is affected by factors other than environmental events, as is explained later. (Castles 2003) mentioned that we would not be able to identify a person as an environmental migrant as he will not migrate only because of environmental reasons. The International Organization for Migration (IOM) has developed a unified concept of "Environmental Migrant" as any person or group who suffered an effective change in the environment or the climate in their lives and chose to leave voluntarily or forced, searching for better living conditions and stability for the short or long term, even internally or internationally (IOM 2019).

From a political point of view, it focused on the legal concept of migrants or refugees and how it can be used. In 1951, the Geneva Convention referred to the refugee/migrant

as someone who leaves his country not to be persecuted because of religion, nationality, race, political opinion, or being a part of a social group. This definition did not have any environmental reason for migration, and this caused two different viewpoints. Some people said the definition should be updated by including the environmental reasons to be able to measure it. Others said that environmental reasons would not be accepted as adequate reasons for migration, which may arise from several factors. Including environmental factors as reasons for migration legally widens the definition of migration and enables the government to withdraw from its political accountability of migration. Hence, governments can negatively use the broad definition of migration to retreat from any responsibilities (Cambrézy 2001). Finally, (Kalin 2008) expressed that it is more important to admit the future existence of environmental migration, regardless of its definition, than be preoccupied with some semantic arguments.

Regarding the discussion of the forced environmental/climate migration issue and the need for climate refuge and asylum, with a particular focus on the EU, (Kallio and Riding 2023) emphasized the importance of incorporating climate-related displacement into EU migration and refugee policies. They highlighted the EU's long-standing commitment to a human-rights-based approach in international affairs and urged expanding this approach to protect individuals displaced by natural disasters. The authors found a massive need for explicit connections to climate-related displacement in the European Commission's proposed migration and asylum pact, highlighting existing frameworks' insufficient effectiveness in addressing climate stress as grounds for seeking refugee status (such as in the 1951 Refugee Convention). Finally, the study criticized current EU policies, particularly the New Pact on Migration and the Green Deal, for failing to address climate-related migration. It mentioned the EU's passive and, at times, aggressive attitude toward climate migrants, highlighting the contrast between the EU's global role in promoting democracy and human rights and its unwillingness to address the issues provided by climate refugees. Lastly, the study confirmed that EU countries' historical responsibilities, disproportionate carbon emissions, and geopolitical consequences ignore climate refugees. It claimed that the EU should be proactive in recognizing the status of climate refugees, connecting its policies with the worldwide need to address climate-induced displacement. In 2023, Australia offered Tuvalu residents a climate change visa and asylum reason. It is the most recent country to provide this kind of visa and this legal right for environmentally vulnerable people. In 2017, New Zealand introduced the same visa for Pacific Island displaced people by climate change, but the plan was dropped. However, the EU should make such agreements despite the deep efforts on the issue.

## 5. The Relationship between Natural Disasters, Climate Change and Migration

The Internal Displacement Monitoring Centre (IDMC) has been analyzing the relationship between natural disasters, climate change, and migration for more than 20 years and found that natural disasters have been the primary reason behind displacement. Data showed 23.7 million people were displaced in 2021 due to natural disasters. In Europe, natural disasters displaced more than 66,000 people in 2017 (IDMC 2017); this number is small compared to the global scale but still important and should be addressed as it still widely affects the European communities. The wildfire in Corsica, France, led to the movement of 10,000 people. The indicators of environmental change in Europe are evident in the increase in temperatures, rainfall variability, changes to the ecosystem, and wildfires. By dividing the European continent, the southern and Mediterranean regions have shrinking water supply, wildfires, and crop production. The northern part is more subject to ecosystem changes and floods. Eastern and central Europe witness an increasing rate of fires and a decline in rainfall, affecting the water supply (IPCC 2014).

The following part presents the previous work on environmental drivers that could affect migration, especially into the EU. These drivers are classified into two categories: fast events and slow events, which could directly or indirectly affect migration (Bardsley and Hugo 2010; Bohra-Mishra et al. 2014).

*5.1. Fast-Onset Environmental Events*

The fast-onset or sudden environmental events such as earthquakes, hurricanes, floods, landslides, and heavy rains are more apparent than the slow-onset ones. The sudden-onset events are more likely to take place for a limited period and in a sudden way, leading to 88.9% of the yearly number of internally displaced people (IDMC 2021). The previous literature found that these environmental fast events mostly lead to a short-distance and temporary displacement (Lonergan 1998; Zickgraf and Perrin 2016), while the long-term and long-distance displacements which lead the migrants to cross the borders, are minimal (McLeman and Gemenne 2018).

The literature has also found that the recurrence of fast-onset events over short periods influences people's livelihoods and migration (Devkota et al. 2017; Kim and Marcouiller 2017). So, the effect of a fast-onset events series should be different or even more potent than just a single effect regardless of its intensity (Berlemann and Steinhardt 2017). Some studies proposed that there could be a link between the recurrence of fast-onset climate events and displacement (e.g., Buchenrieder et al. 2017; Neumann et al. 2015), but the results of these papers did not come from empirical analysis, and due to the lack of needed data, only some of them quantified the effects of the recurrence events (Saldaña-Zorrilla and Sandberg 2009; Bohra-Mishra et al. 2014; de Campos et al. 2017).

Some of the previous literature found that floods and storms do not directly affect migration. This was shown in a case study of Bengali villages; when the households were asked about their migration decisions after the tornado, they said they did not migrate because of the success of governmental policies in dealing with this disaster (Paul 2005). In the U.S., when they also started applying some security schemes to deal with floods during the 1940s, they had lower migration ratios (Boustan et al. (2012)). (Perch-Nielsen et al. 2008) showed that most migration that happens with floods is temporary. This conclusion is also confirmed by (Gray and Mueller 2023) after analyzing the IFPRI household surveys in Bangladesh from 1994 to 2010, which showed that long-term migration is inconsistent with floods.

Earthquakes are one of the most devastating natural disasters. Throughout history, earthquakes have led to the migration of a massive number of people to adapt to their consequences. One of the most damaging earthquakes in history was in Japan in 2011, with a tsunami following it, so people relocated to escape from its effects. The out-migration of Japanese people post the earthquake led to changes in the demographics of some states in Japan. The Hyogo region lost around 80,000 of its population, and the Tohoku region's demographics changed as people aged 65 and above accounted for 40% to 60% of the population (Abe 2014). The large-scale earthquake in West Sumatra in Indonesia also resulted in many migrants. Around 37% of the region's residents chose to migrate due to the earthquake, with more men than women migrating after the earthquake. The people who chose to migrate had a higher level of education than the people who chose to stay (Karimi 2017). In contrast, (Gignoux and Menéndez 2016) argued that a small flow of people migrated from the affected areas after natural disasters in Indonesia due to the projects that took place and prevented people from being subject to extreme poverty or encouraging them to migrate massively.

Nepal is highly prone to natural disasters. In the 2015 earthquake, migration was an adaptive option for people to deal with the earthquake's consequences. Many female workers left Nepal after the earthquake due to the deteriorating economic conditions and the loss of income accompanied by the earthquake (Shakya 2016). However, (Shakya et al. 2022) claimed that the number of work documents issued for international migration after the earthquake decreased by 37.8% in the most destructed areas. This is because many males residing in the destroyed areas remained to assist in the reconstruction, unlike women, who were more likely to migrate. Even though migration might be an adaptive strategy in earthquakes, it can negatively impact developing countries. For instance, Armenia as a small state with a population of three million people only. With the increasing number of labor migrants, the country might face the challenge of low-skilled labor (Demirchyan et al. 2021).

Nonetheless, it should be noted that people will be less likely to migrate when the government and state institutions focus on mitigating the consequences of the earthquakes and developing infrastructure and public health care.

In the EU context, (Armaş et al. 2017) discussed the growing vulnerability of urban regions to earthquake damage in Romania, as seen by recent seismic events, highlighting the necessity for a knowledge of the relationship between spatial dynamics and socioeconomic determinants within cities. The study focused on Bucharest as a seismic hotspot in the EU, creating an overarching vulnerability index for seismic hazards by integrating spatial post-processed socioeconomic data from the 2011 Romanian census with multicriteria analysis and analytical methods in a GIS environment. The interdisciplinary approach used by the study demonstrated the effectiveness of combining spatial multicriteria analysis, analytical methods, and empirical data to create a comprehensive vulnerability index, which provided valuable insights for improving the precision of seismic vulnerability assessments and guiding targeted mitigation efforts in earthquake-prone urban areas.

(Jansen et al. 2017) showed that natural gas extraction in The Netherlands' northern region has resulted in soil subsidence and the advent of earthquakes. This increased the concerns among inhabitants about the safety of their families and the marketability of their homes and potentially encouraged a desire to transfer. This study aimed to investigate the relationship between earthquakes and the intention to relocate through a survey of inhabitants in nine "risk municipalities" in the province of Groningen. Over 19,000 inhabitants were invited, including 811 members of the "Groninger Panel" and 18,436 randomly selected replies; the data showed that age, education, household size, length of residency, and commitment to the location all influence the intention to relocate. The study emphasized the significance of providing people with assistance and psychological care to reduce their intention to relocate and addressing worries about the value and salability of their homes in the face of seismic disasters.

In Italy, the earthquake of Messina-Reggio Calabria, which happened in 1908, is considered one of the most destructive natural disasters in the modern history of the EU. When it happened, many people chose to migrate abroad from southern Italy as the international borders were open, facilitating the migration choice as a response. (Spitzer et al. 2020) showed the effects of this earthquake on international migration from Italy to other EU countries and the US and found that, on average, this disaster positively affected migration. There was heterogeneity in the people's responses, with a more positive relation with agricultural laborers comprising the other share of the labor force. The study showed that this earthquake affected the state's capital and people's sources of income; hence, staying in their home state was no longer an option. It also indicated how much the climatic shocks affect some push factors, leading to migration. The results of the study confirmed that natural disasters and climatic shocks destroy infrastructure, houses, and sources of income, hindering production and trade and destroying capital, as mentioned in previous studies (Baez and Santos 2008; Banerjee 2017; Cai et al. 2016; Gröger and Zylberberg 2016).

*5.2. Slow-Onset Events*

The second category of events is called "slow-onset", characterized by events that are less sudden than the others, such as desertification, droughts, soil erosion, increasing temperature, and increasing rains. Previous conclusions about the impact of these events on migration were very conflicting. Some of them found that these events affected people's displacement, especially interregional ones such as Argentina and Brazil in South America, Iran and Syria in the MENA region, and Ethiopia and the Sahel in Africa (Miyan 2015; Piguet and Laczko 2014). In 1985, Nigeria's drought led to more than one million migrants, either permanently or temporarily (Hammer 2004). (Leighton 2016) found that irregular droughts and desertification in Brazil contributed to the migration of more than 3.4 million people from 1960 to 1980. In Tanzania, (Beegle et al. 2011) showed the effect of rainfall variation on migration stocks over ten years. (Gutmann et al. 2005) also demonstrated that the temperature increase and rainfall variations pushed the migration from North

America to the US from the 1930s to the 1950s. (Afifi and Warner 2008), after studying 172 countries, concluded that all indicators of water shortage, desertification, deforestation, and soil erosion correlate positively or negatively with migration.

(Barrios et al. 2006) was one of the first studies about the effect of droughts on migration. It was a cross-country analysis of 78 countries through the last 30 years. The author noticed that the decrease in rainfall led to rural internal migration, mainly in Sub-Saharan Africa, but it differs in other developing countries. (Munshi 2003) also explored the relationship between droughts, measured by less rainfall, and migration in America. The study found a significant relation as these droughts led to massive migration from Mexico to the U.S. In Ghana (VanderGeest 2011) found that migration is higher from the less environmentally dependent regions and that the droughts measured by less rainfall positively correlate with migration.

Differently, (Gröschl and Steinwachs 2017) found that the drought could increase people's displacement but only in middle-income countries as they have sufficient resources for migration but are not rich enough to have high-security procedures. So, this study found that liquidity constraints are one of the main factors that affect the relationship between environmental change and migration. Droughts can also lead to an increase in temporary migration, especially in rural ones, but without any effect on international migration, even with a reduction. This conclusion was also confirmed by (Kniveton et al. 2008) after analyzing the climate variability and migration relationship in the U.S. from the Mexican regions, specifically Zacatecas and Durango, from 1951 to 1991. Other studies mentioned that droughts could lead to a marginal increase or even decrease in international migration while increasing it internally (Kniveton et al. 2008). The kind of climate event could have a contrasting impact, such as in Mali when there was a drought in the mid-1980s, resulting in the long-distance migration decreasing because of insufficient resources (Findley 1994). The same conclusion was confirmed by (Smith 1996) after the drought in Bangladesh in 1994, which showed that very few people, around 1%, decided to migrate internationally because of the drought.

For the temperature, some studies, (Rose 2001) and (Dillon et al. 2011) in Nigeria, tried to investigate migration as a response strategy before by using rainfall variation and after by using temperature variation as a slow-onset scenario. The authors found a positive relationship between rainfall and temperature variation with men's migration, as there was at least one male migrant in each family due to these changes. (Cattaneo and Peri 2016) also demonstrated a positive relationship between temperature increase and migration in more developing countries.

(C. Cattaneo et al. 2015) investigated the effects of climate change, considering the warming trends recorded by various countries, on agricultural productivity and the subsequent effects on rural income dynamics. The study, which spanned 1960 to 2000 and included 116 countries, examined how different warming patterns affected migration probabilities, either outside the country or from rural to urban areas. The findings indicated that increasing temperatures were associated with increased migration rates to cities and other countries, particularly in middle-income economies with a lower migration risk than low-income countries, probably due to liquidity limitations. In the face of global warming, migration has emerged as a necessary adaptation mechanism in middle-income countries, directing structural changes and increasing revenue per worker.

Rising sea levels are considered a substantial environmental change that could lead to migration or the full sinking of some cities with their population, especially in countries without prepared infrastructures. For instance, the rise of sea level occurred on the coastal American Chesapeake Island in the 19th century and led to a massive displacement of the population from the island at the beginning of the 20th century (Gibbons and Nicholls 2006). The rise in sea level can be measured and projected easily, especially using the GIS (geographical information system), so it is manageable to count the people with high vulnerability and who live in low coastal places. (McGranahan et al. 2007) identified the low-elevation coastal places, which are zones located on the coast with a height of fewer than 10 m. How-

ever, these places have only two percent of the land, with more than 600 million people, most of whom are neglected in the most developing countries (Anthoff et al. 2006).

(McGranahan et al. 2007; Anthoff et al. 2006) mentioned that due to the emerging issue of the global rise in sea levels and its consequences, such as soil salinization, the destruction of ports and coastal roads, interference of salt and fresh water, and territorial losses, people are more vulnerable to fast-onset events with coastal ecosystem degradation. This degradation hinders the ability to deal with sudden events such as tsunamis and storms, leading to less protection, so people will be more and more likely to migrate (McGranahan et al. 2007).

Salinization of the soil is a critical issue that should be tackled because it could reduce coastal crops, hinder seed development, lead to less land productivity, and affect water security due to water contamination. As a result, people become more likely to migrate (Hassani et al. 2021). Other factors affect people's survival in a particular area, such as the melting of ice, land and forest degradation, and the shortfall of biodiversity. Therefore, people shift to regions with better circumstances and better chances of human survival (Le Ha et al. 2021). Besides, some of the sudden-onset events might be affected by the slow-onset events. For instance, slow-onset actions might affect certain sudden events' severity, impact, and regularity (Zickgraf 2021).

In the EU, between 1985 and 2006, (Bonasia and Napolitano 2012) looked at the factors influencing interregional mobility for both unskilled and skilled migrants in Italy. In addition to the usual factors of the Harris and Todaro model, the study considered carbon dioxide emissions as environmental change as one of the potential influencers. The research, carried out using a dynamic two-step panel generalized technique of moments, demonstrated that the standard model may need to be revised. The authors emphasized the importance of externalities, underlining the significance of including broader quality-of-life metrics as explanatory variables. The study found that these non-economic elements and some economic ones, such as housing prices, were possible drivers for migration.

With the continuous presence of slow-onset events, studies are necessary to determine how their impacts and consequences are shaped by checking their drivers and triggers. Unquestionably, natural disasters and climate change adaptation strategies, catastrophe reduction funds, and people's level of vulnerability are all factors that shape the impact of slow-onset events. Therefore, some precautions should be considered to limit the effect of slow-onset actions (Thomas et al. 2020).

To sum up, the literature showed that fast-onset events are more sudden, which is why they mostly lead to internal, temporary, and forced migration. In contrast, the slow-onset events that happen gradually mostly lead to permanent, voluntary mobility, which could be internal or international or even no migration at all.

## 6. Mediating Factors and Environmental Shock Migration

Migration is not a unique phenomenon occurring solely due to natural disasters and climate change. There are mediating factors between natural disasters or climatic shocks and migration. Hence, natural disasters and climatic shocks might influence economic, political, and social factors, eventually leading to migration. Accordingly, there is not necessarily a direct relationship between environmental shocks and migration, as mediating variables can be influential.

On the other side, these economic, political, and social factors could influence natural and climatic shock factors. For example, the effect of environmental change in wealthy and democratic countries differs from its effect on developing countries. (Sen 1982) showed that the effect of climate change is higher in developing countries that do not have enough resources and are characterized by many other political or social issues. In addition, the inability to move, even during shocks because of the insufficiency of other resources, leads to a "trapped population phenomenon". Trapped populations cannot migrate and are one of the most exposed populations to hazards (Findley 1994; Black et al. 2013). The immobility accompanied by the trapped population emerges from factors such as limited

capital, minimum financial resources, and low income (Bryan et al. 2014). Previous studies were able to group the leading causes behind people's choosing not to migrate. First, people's low incomes force them not to move (Blaikie et al. 2003). Second, people might need to comprehend the risk (Morrow 2009). Third, culture can influence people's choice of non-migration and their feeling of belonging to their society (Douglas and Wildavsky 1982; Beck 1992). Fourth, psychological factors and people's actions are essential to immobility (Krämer 2014).

The previous arguments clarified the critical presence of multi-causality, which implies that migration could have an indirect relationship with these shocks and many different factors in between them. Understanding the other possible factors and the complexity of natural disasters and climate change leads to knowing the correct variables to measure the phenomenon with higher certainty levels.

Economic factors are viewed as the primary causes of migration. According to (Lilleør and Broeck 2011), two economic factors boost migration: income fluctuations across time and income disparities between the home and host states. For instance, (Marchiori et al. 2012) demonstrated that governments in the sub-Sahara decrease the income of residents of rural areas to encourage their migration from rural to urban regions. (Feng et al. 2010) also demonstrated a relationship between environmental change and farm income through the link between climate change and crop yields.

Some studies proved that "income changes" are a significant factor affected by environmental change and, therefore, simulate migration. Being at risk of income changes in the future as a household will lead to migration to avoid this uncertainty. (Dillon et al. 2011) found that income change risk because of climate change led to internal male migration in Nigeria. (Marchiori et al. 2015) explored the relationship between income changes because of environmental change and migration in Sub-Saharan Africa and found only a marginal effect of this factor on migration in these countries. Accordingly, literature in developed countries such as the EU focused on studying how these risks are managed before their occurrence. In addition, studies started to tackle the actions taken after the occurrence of these risks, mainly the actions of households as a way to cope with the risks and to protect their spending (Alderman and Paxson 1992; Morduch 1995; Townsend 1994; Dercon 2005).

Developing regions and small municipalities are the most vulnerable to natural disasters and climate change, not just because they mainly depend on agricultural activity but also because they do not have enough resources to deal with or adapt to environmental change. A case study of India as a developing country (Dallmann and Millock 2017) showed that the regions with higher dependence on agriculture have more internal migrations when frequent droughts hit them. (Cai et al. 2016) also investigated the relationship between climate change and agricultural countries and found that temperature change led to more migration in these countries. (Beine and Parsons 2014) found that developing countries, the most vulnerable places to climate change, have a higher level of internal migration when affected by a climatic shock. The previous conclusion is explored firstly by (Ezra and Kiros 2001), who found a consequential relationship between people's migration and their exposure to food shortages in their places.

Liquidity constraints are essential economic factors that arise when citizens need more money or savings to afford migration. People with less liquidity and high poverty levels are less likely to migrate than others with high liquidity (Tiwari and Winters 2019). Some studies perceived migration as a type of investment, especially if it takes place after income shocks, considering that migration leads to better income. The liquidity constraints factor considerably affects migration decisions (Kniveton et al. 2008; Bryan et al. 2014; Cattaneo and Peri 2016). (Borger 2010) also perceived liquidity constraints as the inability of migrants to borrow in return for upcoming earnings. Therefore, such liquidity constraints hinder legal and professionally assisted migration. The usual dilemma starts after a natural shock: the incentive to migrate and the lack of required resources to afford migration. As a result, citizens who cannot afford to move have to deal with the consequences of these shocks

(U.K. Government Office for Science 2011; Black et al. 2011). However, it depends on which factor is more dominant: the liquidity constraints or the incentive to migrate.

Natural disasters, especially fast-onset ones, lead to higher liquidity constraints (Kniveton et al. 2008; Bryan et al. 2014; Cattaneo and Peri 2016). Considering the comparison of the household between the push and pull factors of migration and the needed resources to do it, which is different from one household to another. It means that low-income people will always be under two main hazards: the hazard of environmental change if they live in a climate-vulnerable place and the hazard of being unable to move if they do not have adequate resources (U.K. Government Office for Science 2011; Black et al. 2011).

(Jayachandran 2006; Gray and Mueller 2023; Mueller et al. 2014; Mastrorillo et al. 2016) showed that low-income households are the first category to move because of environmental change. When a climate shock hits low-income households, they choose the simple and fast types of migration, which could be a short distance or for a short time, and this is called "Survival Migration" (Kleemans 2015). Conversely, wealthy households behave differently as they choose long-term or international migration, especially as they can afford it and prefer profitable and more stable migration. (Kleemans 2015) found that international migration is four times more costly than internal migration; hence, low-income families need help to afford it. The author also found that the two types of migration, survival or profit, do not complete each other but are alternatives, which means that if the household chooses one, it is less likely to choose the other.

Political factors also directly relate to the decision to migrate. Without political rights or if the country is involved in political disruption or conflicts, citizens migrate to seek better living conditions and more security (Czaika and Reinprecht 2022). Some studies have proved that droughts and floods have a historically direct relationship with conflict (Homer-Dixon 1991; Bai and Kung 2011). Environmental change and its accompanying changes in temperature, food insecurity, rainfall variations, or crop production strongly contribute to the emergence of conflicts (Bai and Kung 2011). According to (Bai and Kung 2011), environmental change was one of the primary causes of the historical Sino-nomadic conflict in China. The relationship between environmental change, migration, and conflict could also be indirect by the change in income or income growth, as indicated by the studies of (Sen 1982; Miguel et al. 2004; Dell et al. 2009; Barrios et al. 2010; Burke et al. 2013), which found that environmental change did not motivate migration in more prosperous and democratic countries and mentioned that most of the moving mainly happened because of political reasons, not climatic ones. (Miguel et al. 2004; Reuveny 2008; Buhaug 2010; Ciccone 2011; Hsiang et al. 2013) found that in the future, environmental change may lead to more conflict and migration. This may happen when the country needs more substantial organizations with frail responses to natural disasters and climate change (Bernauer et al. 2012). (Kelley et al. 2015) expected that the extended drought in large parts of Syria was one of the main reasons that enlarged the migration phenomenon in parallel with the political reasons.

Gender is another essential aspect (Chindarkar 2012). Previous studies showed that women are more vulnerable to climate change than men for many reasons, such as the inequality gap, limited resources access, and labor gap. (Becerra-Valbuena and Millock 2021) investigated the effect of droughts on gender migration in Malawi and found that the climatic shocks measured by droughts led to a slight decrease in marriage migration among young girls while increasing it among women of older ages. Women of older ages migrate for marriage to get over the implications of the severe droughts. The study also found an internal increase in labor migration among young men because of this climate change, while it is the same for working girls but with a lower level of increase. (Gray and Mueller 2012) found that men's labor migration increased after the drought with high rates to raise their income in Ethiopia. This phenomenon of labor migration acts as a coping mechanism to deal with the implications of droughts. In contrast, the marriage mobility of Ethiopian women decreased after droughts. Such a decline is compatible with avoiding high expenses related to marriage migration.

In the EU, the literature dived into the pressing challenges of migration and environmental change, which have caused notable anxiety and nervousness among European citizens. (Bettini et al. 2021) examined the remarkable distinction between these anxieties, highlighting their political significance at Sardinia in Italy. The study investigated the dynamics of fear generation, mobilization, and contestation through focus groups and interviews with Sardinian municipal officials. However, instead of being regarded as security problems, these challenges become more related to socioeconomic matters such as austerity, depopulation, economic decline, and rural-urban dynamics. The study concluded the consequences of this dramatic difference for broader debates on climate and migration, revealing how these concerns overlap at the local level.

(Henning et al. 2022) examined the complex interaction of economic, conflict, and environmental elements impacting public acceptance of immigrants in Austria as a receiving country. The study manipulated hypothetical migration scenarios from Chad, varying causes such as violent conflicts, climate change, local environmental degradation, and economic prospects, using an online survey experiment with 686 student participants from the University of Innsbruck in Austria. Surprisingly, a sizable minority (25%) would be against conflict migrants. The study also highlighted the factors influencing acceptance, such as respondents' perceptions of links between welfare, crime rates, job possibilities, and immigrant presence. Gender and political affiliation also have a role, with right-wing preferences connected with lower levels of acceptance. The article added critical insights to the discussion on migrant acceptance, especially in light of political and societal divisiveness. The analysis was enhanced by detailed demographic information on the subject pool, such as age, gender, origin, political affiliations, and attitudes toward immigration. The authors emphasized the significance of understanding immigration sentiments among today's students, who will affect future immigration policies in response to the changing climate migration geography.

(Kwiliński et al. 2022) also dived into the varied influences of social, economic, ecological, and political drivers on regional and worldwide migration by exploring recent patterns in international migration and the many perspectives surrounding global population mobility. The primary focus is on examining and comparing the deep causal links between international migration and the economic, ecological, and sociopolitical components of the growth of EU members. The detailed analysis in the discussion and conclusion compared structural similarities and variations among EU nations, shedding light on their responsiveness to economic conditions and changes. The study emphasized the importance of having well-managed migration to contribute to important UN Agenda 2030 goals such as decent work and economic growth, reduced inequality, climate action, and the promotion of peace, justice, and strong institutions.

To conclude, there is not necessarily a direct relationship between environmental shocks and migration; economic, political, and social factors can influence or be influenced by environmental shocks and subsequently affect migration. Economic factors such as liquidity constraints, income level, and income variability are critical for migration as a response to environmental shocks. They could encourage people to migrate for a better living after the shock or prevent them from relocating if they are insufficient.

## 7. Types of Environmental Shock Migration

Several types and forms of environmental shock migration must be examined, such as internal or international, long or short, temporary or permanent, rural or urban, and forced or voluntary. The form depends on the number of people who chose to move, the region, the migration duration, and the context. (Xiang and Nyberg Sørensen 2021) argued that shock migration could be classified into five types. The first type is "reaction mobility", a sudden response to severe shocks. The second type is "reaction immobility". Just as people might choose to move with the occurrence of shocks, others might choose not to move. The third type is "survival mobility", which is essential for people losing their sources of income during shocks and mobility limitations. The fourth type is "limbo mobility", which

occurs when people decide to migrate but do not know their destinations or how they will reach them or are rejected by other communities. The fifth type is "substitution mobility", which occurs mainly during lockdowns and is regarded as people moving instead of others.

The nature of migration, either temporary or permanent, could be related to the nature of the shock itself as a slow-onset or fast-onset event. So, climatic events, regarded as slow-onset events such as desertification, rising sea levels, and riverbank erosion, mostly lead to permanent migration (Gutmann et al. 2005; Gibbons and Nicholls 2006). Households living close to riverbanks often experience the loss of homestead and agricultural land, which reduces their production or employment opportunities and threatens their livelihood security (Alam et al. 2016). People living in coastal places are also particularly exposed to such permanent natural disasters as riverbank erosion and find migration a viable coping strategy (Poncelet et al. 2010; Brammer 2014; Kabir et al. 2018; Mollah and Ferdaush 2015). In contrast, fast-onset events or natural disasters such as hurricanes, earthquakes, cyclones, or storms mostly lead to temporary migration (Perch-Nielsen et al. 2008; Bohra-Mishra et al. 2014; McLeman and Gemenne 2018). Some environmental events, such as drought, lead to both types of migration depending on other factors such as geographical location.

States subject to temporary environmental shocks most likely experience them repeatedly with different severity, time, and recurrence levels. For instance, Bangladesh is subject to repetitive temporary shocks manifest in cyclones, storms, and high winds, which average every three years. (Dasgupta et al. 2010; Mallick and Etzold 2015). Bhola in 1970, Sidr in 2007, Aila in 2009, and Komen in 2015 are examples of destructive cyclones in Bangladesh in the last years. These temporary shocks led to a permanent migration of millions of Bengalis due to the destruction of the infrastructure, elimination of sources of income, scarcity of resources, and lack of social protection concurrently occurring with the shocks (Poncelet et al. 2010; Mallick et al. 2017)

The kind of migration, whether it is internal or international, is also critical. (Gray and Bilsborrow 2013) demonstrated the existence of a negative relationship between internal/international migration and the annual rainfall mean in Ecuador. The study also showed that crop variation does not affect international migration but increases internal migration. The conclusion was that international migration is the most challenging type, which is why it is the last to occur due to natural disasters and climate change. The author also found an increased probability of international migration for land-owner households, which confirms the previous assumptions about the role of wealth and how much international migration costs.

The resulting natural disasters and climatic shocks lead to the migration of citizens of developing states to developed states such as the United States of America (USA), Canada, France, Australia, the United Kingdom (UK), and Germany into the EU. However, it has been apparent that most of the migrants to these states are educated people; this can be attributed to an assumption that highly educated people can afford migration as they probably have a high income. It became evident that natural disasters and climatic shocks might lead to international migration with some considerations (Drabo and Mbaye 2015; Munshi 2003; Marchiori et al. 2012).

By studying environmental change and migration, the economic and geographic channels lead to internal migration and, subsequently, to international migration. According to (Marchiori et al. 2012), the economic aspect and its link to international migration is a phenomenon known as the "economic geography channel". It aims to prove that climate changes lead to internal migration, contributing to lower income in urban areas and eventually forcing people to cross borders. The study argued that even though economic conditions are highly influential, climate changes can still independently affect international migration; this phenomenon is known as the "amenity channel". It shows that regardless of the economic geography channel, climate changes might force people to migrate internationally due to increasing temperature, deteriorating general environmental quality, high death rates, and the spread of diseases. Therefore, analyzing the amenity effect will show a direct relationship between climate change and international migration.

Previous studies such as (Beine and Parsons 2014; Beine and Parsons 2017; Cattaneo and Peri 2016) examined the relationship between internal migration and environmental change. These studies rely on calculating a country's ratio of urban to the total population, widely known as the "urbanization rate". The reason behind using the urbanization rate is the lack of sufficient information measuring the number of internal migrants moving from rural to urban areas over time. However, things are different regarding the nature of the environmental shocks, their location, and their impact on internal migration. (Bohra-Mishra et al. 2014) analyzed the impact of long-term and short-term natural shocks on migration in Indonesia. The study showed that sudden events led to temporary internal migration, while slow events, such as temperature changes and increasing rainfall, led to permanent internal migration. The study concluded that developing states are highly affected by the short-term shocks that lead to an accelerating temporary urbanization rate. Meanwhile, middle-income states are affected primarily by long-term shocks, and some studies relate this result to liquidity factors and migration costs (Cattaneo and Peri 2016). The following Table 1 is a summary of the environmental shock migration types extracted from previous studies.

**Table 1.** Summary of environmental shock migration types from previous studies.

| Temporary | Permanent | International | Internal | Forced | Voluntary |
|---|---|---|---|---|---|
| The fast-onset shocks such as floods, storms, or earthquakes lead to temporary migration. | Slow-onset climatic events such as desertification and sea level rise lead to permanent climate-induced migration in most cases. | International migration is the most challenging due to its cost. Some studies found that environmental shocks have a positive effect on international migration while others found that they have no effect or even a negative one. | The most possible type of happening. Some Studies found that environmental shocks, even slow or fast, mostly increase internal migration. | Fast-time onset shocks mostly lead to forced migration internally. | Gradual climate change and Slow-onset shocks are more connected to this kind of migration. |

Source: prepared by the author.

## 8. Conclusions

To conclude, this paper aims to assess the relationship between environmental change and migration, especially into the EU. To prove such a relationship, it was essential to explore the connection between climate change, natural disasters, and the subsequent environmental shock migration, defined as the abrupt movement of people due to a climate shock or natural disaster. The term "Environmental Shock" primarily refers to the large-scale effect of the environmental event with a definite subsequent effect on the migration. Connecting the environmental shock with the consequent migration resulted in the paper's term: "Environmental shock migration". The Internal Displacement Monitoring Center (IDMC)—which analyzes the natural disasters, climate change, and migration relationship—showed that in 2017 natural disasters displaced more than 66,000 people in Europe. In 2021, 23.7 million people were also displaced globally due to natural disasters.

Throughout this study, it became evident that several forms of environmental shock migration primarily depend upon the context of migration, the duration, the number of migrants, and the region. These factors formulate several types of environmental shock migration: long, short, internal, international, forced, voluntary, temporary, and permanent. To better understand the types of environmental shock migration, it is essential to categorize the types of environmental events. The study has shown two types of climatic events: fast-onset events and slow-onset events. Slow-onset climatic events, such as desertification and rising sea levels, mostly lead to permanent migration (Gutmann et al. 2005; Gibbons and Nicholls 2006). In contrast, fast-onset events or natural disasters such as hurricanes, earthquakes, or storms lead to temporary migration (Perch-Nielsen et al. 2008; Bohra-Mishra et al. 2014; McLeman and Gemenne 2018). Analyzing fast-onset events is much more precise than slow-onset events. Fast events are more evident, making their implications easier to analyze. In Italy, after the earthquake of Messina-Reggio Calabria in 1908, many people chose to migrate from southern Italy, especially since the international

borders were open. One study (Spitzer et al. 2020) showed that this disaster led to the sudden movement of people internationally for the long term without return in some cases.

The impact of environmental change and migration into the EU was also evident. Environmental change with the resulting natural disasters and climatic shocks leads to the migration of citizens residing in developing states (Drabo and Mbaye 2015; Munshi 2003; Marchiori et al. 2012). Their destination is chiefly in countries such as the United States (US), Canada, Australia, or the EU, such as the United Kingdom (UK), France, and Germany. These international migrants are highly educated in their developing countries and are most likely to have high incomes. Thus, they might be able to afford international migration to the EU.

Natural disasters and climatic shocks might not directly lead to migration as there are some mediating variables that could affect it. Economic factors, such as income disparities and income fluctuations between host and home countries, are one of these primary mediating variables (Lilleør and Broeck 2011). Remittances are also important as they encourage people to migrate to increase their families' income level in their origin state and act as a frontier against climatic shocks. Liquidity constraints widely affect the decision to migrate. After environmental shocks, people with high liquidity are more encouraged to migrate than those with low liquidity. Political and social factors are also important mediating variables. Political factors such as the degree of political rights in migrants' home countries, political stability, and political conflicts might affect migration decisions. Social factors such as gender disparities and women's vulnerability to climatic shocks might also impact migration.

Recently, data on environmental shock migration became more available with increasing academic work at the micro and macro levels. The results of this work are diverse as they used various approaches and methodologies. Although some previous studies found an apparent effect of natural or climatic shocks on people's migration, others had different outcomes. Despite all recent work, we still need a harmonized theoretical and empirical methodology to illustrate the relationship between environmental shocks and migration, and we still have some gaps to address, which are possible for future studies, as summarized in the following Table 2.

**Table 2.** Summary of the literature gaps in the EU and needed future work.

| Research Gap (Issue) | | Future Work |
|---|---|---|
| 1. | The direct relationship between climate change and migration | More empirical work without passing through the channels of this relation (qualitative work) |
| 2. | Gender and environmentally induced migration | More empirical work includes gender and environmentally induced migration especially into/to EU countries. |
| 3. | Multi-causality issue | More empirical analysis and case studies into the EU to investigate the determinates of Environmentally induced migration decisions and their channels, specifically in the case of immobility. |
| 4. | Heterogeneity issue | More analysis into EU countries to explain the heterogeneity of migration responses in terms of other possible causing factors such as gender, wealth, human capital, financial capital, health, and age. |
| 5. | Endogeneity issue | Deep empirical work and case studies into the EU countries considering the self-selection base and trying to deal with the endogeneity problem. |
| 6. | Urbanization level (rural and urban areas) | More micro-level academic work to determine the case of households of specific EU cities/countries in terms of agricultural/rural and non-agriculture/ urban places. |
| 7. | The geographical gap | More diverse spatial studies explain Environmentally induced migration as a kind of mobility from one place to another, especially the neglected small, inner, and developing areas in EU countries. |

**Table 2.** *Cont.*

| Research Gap (Issue) | Future Work |
| --- | --- |
| 8. The term gap | More case studies into the EU to include the effect of more adverse environmental change with increased frequency over time and how households respond to incremental shocks. |
| 9. Climate-induced migration projections | The lack of data and models in the EU motivates us to enhance future estimations of environmental-induced migrants and their characteristics with a higher level of certainty. |

Source: Prepared by the author.

To deal with the environmental shock migration phenomenon effectively, it needs to be analyzed comprehensively. Governmental policies directed towards environmental shock migration are necessary. Policies must also be directed towards slow-onset events, not only the fast-onset ones. Yet, slow-onset events are usually neglected even though they highly contribute to massive waves of migration. Governments also need to focus on adaptive strategies post shocks other than migration by enhancing their facilities and providing social security and insurance schemes; this could mitigate the implications of natural disasters and reduce migration.

### 9. Policy Recommendations for the EU

Before moving forward, the EU should create precise phrases and implications. It should indicate its position on concepts related to climate change and environmental degradation, as well as the consequences of these notions. The EU should encourage collecting data on the impact of environmental changes on migration and displacement, engaging with regions and countries facing mobility constraints because of these issues, and including different approaches, such as the IDMC approach for data on cross-border displacement induced by disasters. The EU should also attempt to include mobility considerations in various EU policies, initiatives, and programs relating to climate change and natural disasters.

The EU should assist in creating and implementing free movement agreements, such as the "IGAD Protocol on Free Movement of Persons". These agreements should be highlighted to establish migration channels or humanitarian corridors for individuals affected by climate change or gradual and unexpected disasters. In addition to proposing a proactive approach that anticipates and resolves future hazards, minimizes vulnerability, and builds resilience for migration, it serves as a voluntary adaptation method. This involves supporting the construction of labor migration routes in Europe and internationally, particularly for those in need to relocate permanently.

Finally, the call for an EU asylum and migration strategy that addresses environmental migration and displacement became necessary by providing the needed technical and financial assistance to national-level projects, particularly within the EU.

**Funding:** This research received no external funding.

**Institutional Review Board Statement:** Not applicable.

**Informed Consent Statement:** Not applicable.

**Data Availability Statement:** Not applicable.

**Conflicts of Interest:** The authors declare no conflicts of interest.

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
