# Peer review of "Does Environmental Change Affect Migration Especially into the EU?"

_socsci, doi:10.3390/socsci13030160_

Round 1
Reviewer 1 Report
Comments and Suggestions for Authors
This article looks at migration due to environmental stresses. This is an interesting topic that will gain in importance as conditions change in various regions of the world. While I think the topic is an important discussion, I do not understand the author's organization or selection of academic literature. Pederson's typology of migration is a seminal organizing typology of migration, one of which is due to environmental stress. When I have reviewed migration, I started with that article. In addition, the article's title states that it will be reviewing reasons for migration to the EU. There is very little literature discussing migration to the EU, which should be the point of departure for a literature review on migration to the EU. I understand that you are attempting to demonstrate the importance of environmentally motivated migration to the EU, but the literature you cited does not make that connection. You cite literature which include source regions that have not been significant contributors to migration to the EU. If this is your interest, I would suggest you review the EU migration literature to determine the factors that influence migration to the region and then provide evidence for the significance of migration due to environmental stressors.
Also, I struggled with the readability of the article due to your syntax and word choices. I would strongly suggest you proofread your article and maybe have someone else review your article for readability, word use and syntax.
Thank you for the opportunity to review your article. I learned a lot about environmental migration. Best of luck in your future research.
Comments on the Quality of English LanguageI struggled with the readability of the paper due to your word choices, syntax and grammar. I would also make sure you explain terms you introduce in the paper. For example, you talk about international migration from the Great Plains counties to the United States. Where are the Great Plains counties? I was unsure if you are talking about the Great Plains of the United States or some other country. I would strongly suggest you proofread your article and have someone else review the article for readability as it relates to syntax, grammar, usage, and word choice.
Author Response
Thank you so much for your time and valuable comments. I sent the paper to a native speaker to proofread it as you recommended and edited the unclear parts you mentioned.
Thanks again
This article looks at migration due to environmental stresses. This is an interesting topic that will gain in importance as conditions change in various regions of the world.
While I think the topic is an important discussion, I do not understand the author's organization or selection of academic literature. Pederson's typology of migration is a seminal organizing typology of migration, one of which is due to environmental stress. When I have reviewed migration, I started with that article.
Response: Thank you so much for your time and valuable comments on my paper, I really appreciate that. For the organization of the literature, I started historically from the first studies conducted about the topic, which were more demographic. Then, I tried as much as possible to set the studies about the phenomena in ascending order of year, including various regions, not only the EU.
In addition, the article's title states that it will be reviewing reasons for migration to the EU. There is very little literature discussing migration to the EU, which should be the point of departure for a literature review on migration to the EU.
Response: Thank you so much for this comment; the article aims to study the phenomena "into" the EU, which means to the EU or internally among the European countries themselves. I agree with you that the article needed more literature about the EU, and I added the most relative studies I found in the edited version, so thanks for the comment again.
I understand that you are attempting to demonstrate the importance of environmentally motivated migration to the EU, but the literature you cited does not make that connection. You cite literature which include source regions that have not been significant contributors to migration to the EU. If this is your interest, I would suggest you review the EU migration literature to determine the factors that influence migration to the region and then provide evidence for the significance of migration due to environmental stressors.
Response: Regarding this comment, I changed the title to "Does Environmental change affect migration, especially into the EU?". So, the reader should understand that the paper generally demonstrates the phenomena with some focus on the EU. Moreover, as I mentioned before, I chose "into" the EU because I had a few case studies with the movement of people among the European countries so that they could be host countries but also origins.
Also, I struggled with the readability of the article due to your syntax and word choices. I would strongly suggest you proofread your article and maybe have someone else review your article for readability, word use and syntax.
Response: I am sorry to hear that; as I am not a native English speaker, I did a proofreading with a native one as you suggested, so thanks for letting me know.
Thank you for the opportunity to review your article. I learned a lot about environmental migration. Best of luck in your future research.
Response: Having you as a reviewer with these critical comments is a pleasure. I thank you for your time and suggestions.
I struggled with the readability of the paper due to your word choices, syntax and grammar. I would also make sure you explain terms you introduce in the paper. For example, you talk about international migration from the Great Plains counties to the United States. Where are the Great Plains counties? I was unsure if you are talking about the Great Plains of the United States or some other country. I would strongly suggest you proofread your article and have someone else review the article for readability as it relates to syntax, grammar, usage, and word choice.
Response: I am sorry again to hear that; as I am not a native English speaker, I did a proofreading with a native one, as you suggested. And for this part of the Great Plains counties, I edited it. So, thanks for letting me know that it's unclear.
Reviewer 2 Report
Comments and Suggestions for Authors
Content and contribution to scholarship: This is a comprehensive review of the state of the literature on how to distinguish the different natural phenomena driving people to move internally and transnationally, temporarily and to seek out more permanent forms of stay. Whereas the authors identify the notion of economic shock migration, the mainstay of the paper --and its strength lies more in the hypothesis that while slow-onset natural disasters trigger temporary migration, while environmental change in a longer-term leads to people moving with a more permanent vision in mind. This dichotomy , which in reality must be less absolute than what the authors might have in mind, is then resolved by the 'mitigating factors', which are socio-economic triggers, as in liquidity, remittances, and which might lead a temporary migrant to stay or to return, regardless of the environmental factor that lead to her decision to move, in the first place. While these second-order factors listed by the authors make sense, there is too little emphasis on host state migration policies which might as well grant temporary admission only for war and conflict, but disregard an intersectionality of these towards environmental factors, such that there is no admission for stay, or only a limited one etc. Here, I find that the EU migration policy, as announced in the title would play a role, however, the EU is not figuring anywhere as a case study, which leads to the conclusion that it be better dropped from the title and abstract altogether. The case studies listed and tentatively compared (more of a cross-comparison would be desirable to illustrate the slow/perm., rapid/temporary conundrum) deal with countries of the Global South or intra-South migration; not so clear. Towards the end of the paper, it remains yet unclear, which directionality of migratory movement is affected and how by which type of environmental crisis: return migration (this is well written, although the authors need to take care that it's called 'return and NOT returned' migration), internal displacement, international migrations? These sections, are clearly less well argued and researched than the environmental trigger ones. Also, please have a look at Walter Kälin (IDP) and Elisa Fornale (environmental migration). Otherwise, the literature review is very comprehensive. The conclusion is fine, but I would like to see an in-depth discussion of the arguments why the categorization identified by the author/s or why the choice of term 'shock' migration is useful, innovative, closes a research gap and where there might be grey areas? are there any situations or country cases, where sudden environmental change (volcano, earthquake) led to permanent emigration, with or without return? Perhaps, the author/s could invest the time in conceiving their own graph or table illustrating these categories and adding where the dynamic elements (of change/fluctuation/permeability) are.
Structure: The paper is too long for a standard journal article, since the arguments needed for building the above-mentioned research hypothesis are sometimes hidden in long discussions of literature that lead to not much new insights for the author/s argument-building. The abstract fails to reflect any of the interesting research hypothesis. The body of the text does not explain to the reader why the new category of environmental shock migration becomes necessary in the field of environmental migration studies: what does it add to the field which is lacking for now? The section on methodology (7) reads well, but should come right after the introduction. At the same time, it would probably be equally useful to set the stage by first introducing the two notions of environmental migration, and only subsequently discuss the different types of migrations affected by the environmental change.
The paper suffers from throwing knowledge at the reader in a rather undigested manner. Hence, the different sections hardly connect to one another and the line of argument is not readily available. A lot of the hard research work undertaken by the author/s remains hidden away in paragraphs that hardly serve to build the argument. It would be good to summarize the key findings after each section ends, to ease the transition into the next section.
Style: this article appears to have been written by non-native English scholars and the language has not yet been edited by an able language editor. This shows as there are several, even if unintended, unacceptable uses of terms made: 'undeveloped countries', instead of countries of the Global South or developing countries, and very early on in the text 'countries suffering from migrants'. Such terminological uses are derogatory and discriminatory and bear a reputational cost for the authors and this journal. Hence, the text must be extremely carefully reviewed by a native English language speaker or editor against such phrasings, before it can be published. Finally, there is one ethical issue, which needs to be worked out before the article can be published: all of the graphs included in this piece are copy/pasted or taken from the Internal Displacement Monitoring Program, the question is, whether this has happened with their full consent? Have the authors obtained an agreement that they can use the IDMP graphs and if so, why is this not mentioned in the 'conflict of interest' disclosure note or the author note? Not being an expert on copyright issues, it seems to me that the copyright has to be clearly established under each graph and I am not sure about the innovative quality of graphs in a paper that are not the author/s' own--this would be different for a conference presentation in powerpoint. To me, this issue needs to be clarified, before the paper can be published.
Comments on the Quality of English Languageas noted under the section 'style'.
Author Response
Thank you so much for your time and valuable comments. I really appreciate your profound remarks and helpful suggestions to improve the paper. Therefore:
1- I added some case studies from the EU.
2- I clarified as possible the mentioned parts, such as the direct migrating movement is affected and how by which type of environmental crisis, by providing case studies.
3- I rewrote the conclusion to have an in-depth discussion of the arguments and why the choice of term 'shock' migration is beneficial, mentioning the grey areas, research gaps, and needed future work regarding the topic.
4- I deleted the IDMC graphs and created two tables to summarize what I have found in two sections, but because I do not want the paper to be as long as you suggest, I did not add more. Also, because the paper is too long, I deleted some parts to keep it balanced.
5- I added the hypothesis to the abstract.
6- As you suggested, I summarized each section and added an intro for the following one.
7- As you suggested, I rearranged the sections by putting methodologies directly after the Intro and background.
8- I had the paper proofread by a native speaker.
Thanks again for your comments. They were very beneficial for me, and I learned a lot.
Please see attached for more information.

Round 2
Reviewer 1 Report
Comments and Suggestions for Authors
I would like to thank the authors for the revised version of the paper. I think the authors have significantly improved the article. I'm still struggling to understand the organization of the article and the selection of literature for the review. I would continue to recommend Petersen's typology as a point of departure for the literature review. Petersen creates a typology that would help the authors organize and situate their research in the academic discourse.
I'm also not sure if the organization of the specific sections generates a logical order for the literature review. You conclude with the "Types of Environmental Shock migration". If I was writing the paper, I would have this at the beginning. I'm also not sure if this title is correct as all of your categories are truly types of environmental shock migration. You hint at this in line 779-780, where you list different types of migration, some of which are included in other sections. I would suggest you rethink your organization for the sections.
As your paper is somewhat lengthy at the moment, I would also suggest you consider shortening or omitting section 6 (Mediating factors) from this paper. If I was writing this paper, I would mention their significance, but not break it out into another section. This is potentially a follow on manuscript to your current project.
In the introduction, you mention that the "movement of people as a response to these environmental changes is known as a 'shock migration'". I'm not sure if this is an oversight or not, but shock migrations could be caused by other stressors, including economic and political. Since you refer to environmental shock migration several times later in the paper, I'm thinking this is more of an oversight as opposed to a conscious decision.
Your writing is much improved from the previous version. Thank you. There are still a few minor errors in the paper. Nothing that significantly impacts the readability of the paper. I would suggest you continue to proofread the paper as you revise it.
Overall, thank you for the opportunity to review you paper. This is a very interesting and relevant topic. Best of luck in your future research.
Comments on the Quality of English LanguageOverall, the paper reads much better than the previous draft. There are a few minor errors, nothing that significantly impacts readability.
Author Response
Thank you so much for the time to review my article.
I am sorry to hear that; I had the chance to read about Petersen's typology of migration, as you suggested in the last report, so thanks for letting me know about it. Petersen's typology is not the best to describe what I have seen in previous studies. Petersen's typology divided migrations into five classes: primitive, impelled, forced, free, and mass. Each class was further subdivided into conservative Migration and innovative Migration. Such typology is based on migration between societies with different levels of culture, peaceful-warlike differences, and differentiation among colonization, immigration, invasion, and conquest. While the typology I used is based on the analysis of socioeconomic systems interactions regarded as a kind of living system within a framework of the Living Systems Theory (LST). Considering migration as a part of a broader socioeconomic system is essential for understanding human migrations and consolidating them into a more comprehensive, productive framework. For example, The initial intentions of migrants may change under the pressure of circumstances; the resulting migration stream also varies accordingly. Other new migration classifications, such as internal-international, mover-stayer, and legal-illegal (regular-irregular) distinctions, are based on the "Boundary" subsystem, which was not included in Petersen's typology.
To sum up, I meant that the article focuses more on the environmental drivers and their relationship with migration in the existence of any other mediating variables, so migration here is a consequence, and I wanted to show it in case studies as it is without any classification. I tried to classify the drivers/types of environmental change that may lead to any migration.
The previous justification takes us to the second point, the organization of the article, and why I ended up with the migration types. Firstly, it was a suggestion from the other reviewer. When I thought about it, I found it logical because, as you mentioned, the migration types have existed in all previous sections since I started the article; that's why I ended up with it as a summary of what I mentioned before. Also, I prioritized the drivers, whether environmental or not, because they are independent and fundamental variables to be analyzed in the paper. Moreover, I summarized and removed from the mediating part, but because it's essential in the article, I couldn't pull it all and only mention its significance. The importance of this part is these variables can totally change the environmental migration relationship or shape it in a specific way, especially since it has been proven through the article that environmental change might not directly lead to migration; that's why I wanted to mention some case studies to prove that.
In the introduction, you mention that the "movement of people as a response to these environmental changes is known as a 'shock migration.' Thank you so much for this comment; I edited it. And definitely, I will continue to proofread the paper as you recommended.
Thank you so much again for your insightful comments.
Reviewer 2 Report
Comments and Suggestions for Authors
The authors have engaged thoroughly with the reviewers' suggestions, such that now, the article features a concise methodologies chapters, has improved much in terms of structure, introduces the key concepts in line with recent literature and has woven into it, several key recent sources (Hoffmann, Henning etc).The authors have also inserted new, well-thought out and well-placed graphs and tables. The article has much improved from the previous version and qualifies as a valuable addition to the environmental migration literature.
Author Response
Thank you so much!
Round 3
Reviewer 1 Report
Comments and Suggestions for Authors
I would like to thank the authors for this version of the paper. I have two main concerns with the paper. First, situating the paper as a study of migration in the EU seems like an afterthought. The authors include a few examples of literature at the end of each section, but do not significantly engage with the example, or provide primary sources from the example to demonstrate its applicability as an example of environmental migration. The authors need to decide if this is a literature review, demonstrating the importance of environmental migration or if it is a study of European migration due to environmental stressors.
Second, I think the paper needs a significant amount of revision. I'm not I understand the separation of sections 5 and 7. They both provide a typology of migration due to environmental change. If I was writing the paper, I would combine the two sections and make one large typology of migration resulting from environmental stressors. This would also help to organize the overall review of the literature.
Thank you for the opportunity to review your research. Best of luck in future projects.
Comments on the Quality of English LanguageThank you for the continued improvement in the quality of your writing. There are a few minor editorial issues throughout the paper. I think another proofread of the paper would help identify the remaining issues.
Author Response
Thank you so much for your comments,
Actually, regarding the EU, I just wanted to give some examples as I was not able to consider it deeply due to the time limit. But I agree with you that I should include more about the EU.
Regarding the typology of migration in sections 5 and 7. You are totally right; the only difference from my side is the reader's view. So, if he is looking primarily at the environmental driver and its impact regardless of the type of migration, it's section 5 (more important in the paper). If the reader is looking primarily at caused migration regardless of the kind of environmental driver, it's section 7. I agree with you it's better to combine or even delete the last one (section 7), but I kept it as recommended by the other reviewer.
Thanks again for your insightful comments.